# Polymer–Metal Interfacial Friction Characteristics under Ultrasonic Plasticizing Conditions: A United-Atom Molecular Dynamics Study

**DOI:** 10.3390/ijms23052829

**Published:** 2022-03-04

**Authors:** Wangqing Wu, Changsheng He, Yuanbao Qiang, Huajian Peng, Mingyong Zhou

**Affiliations:** State Key Laboratory of High Performance Complex Manufacturing, School of Electrical and Mechanical Engineering, Central South University, Lushan South Road 932, Changsha 410083, China; csu-hcs@csu.edu.cn (C.H.); yuanbaoqiang@csu.edu.cn (Y.Q.); huajian_peng@163.com (H.P.)

**Keywords:** interfacial friction, ultrasonic plasticization microinjection molding, polymer–metal interface, contact mechanics, transient temperature, polymer segment rearrangement, energy evolution

## Abstract

Understanding the properties of polymer–metal interfacial friction is critical for accurate prototype design and process control in polymer-based advanced manufacturing. The transient polymer–metal interfacial friction characteristics are investigated using united-atom molecular dynamics in this study, which is under the boundary conditions of single sliding friction (SSF) and reciprocating sliding friction (RSF). It reflects the polymer–metal interaction under the conditions of initial compaction and ultrasonic vibration, so that the heat generation mechanism of ultrasonic plasticization microinjection molding (UPMIM) is explored. The contact mechanics, polymer segment rearrangement, and frictional energy transfer features of polymer–metal interface friction are investigated. The results reveal that, in both SSF and RSF modes, the sliding rate has a considerable impact on the dynamic response of the interfacial friction force, where the amplitude has a response time of about 0.6 ns to the friction. The high frequency movement of the polymer segment caused by dynamic interfacial friction may result in the formation of a new coupled interface. Frictional energy transfer is mainly characterized by dihedral and kinetic energy transitions in polymer chains. Our findings also show that the ultrasonic amplitude has a greater impact on polymer–metal interfacial friction heating than the frequency, as much as it does under ultrasonic plasticizing circumstances on the homogeneous polymer–polymer interface. Even if there are differences in thermophysical properties at the heterointerface, transient heating will still cause heat accumulation at the interface with a temperature difference of around 35 K.

## 1. Introduction

Polymer–metal interfacial friction has been widely exploited in the advanced molding of plastic materials as a typical functional heterogeneous interface behavior. It is critical to comprehend and grasp its behavior to apply it successfully. Plastic pellets, for example, can be successfully conveyed in the plasticizing unit of injection/extrusion molding machines by customizing the interfacial friction between the barrel and the screw [1,2]. With its benefits of rapid heat generation, low energy consumption, and high material usage rate, ultrasonic plasticization microinjection molding (UPMIM) technology has recently become an appealing version of thermoplastic microinjection molding [3,4,5,6,7,8,9,10,11]. Interfacial friction in UPMIM plasticizing units has aroused researchers’ curiosity to understand the quick heat generating mechanism [11,12,13,14,15,16,17]. Michaeli et al. [15] investigated the heating mechanism during ultrasonic plasticization using theoretical analysis and determined that heat generation was mostly due to polymer damping characteristics and polymer–polymer interfacial friction. Li et al. [16] employed finite element modeling to investigate the effect of process factors on the interface heat generation rate, concluding that the effect of ultrasonic amplitude on the transient heat generation of interfacial friction was more significant. We [17] investigated the interfacial friction heating characteristics and mechanisms of the polymer–polymer interface in UPMIM using experimental and numerical simulation methods, and concluded that interfacial friction heating is a transient process that occurs only at the beginning of ultrasonic plasticizing.

To the best of the authors’ knowledge, the open literature has reported the characterization of the frictional temperature evolution and heating rate at the polymer–polymer interface under ultrasonic plasticizing conditions, which partially interprets the friction behavior of the polymer–polymer interface using theoretical analysis, finite element simulation, and experimental methods. However, the molecular mechanisms of interfacial friction heating remain unknown, particularly the dynamic reaction of macromolecules under ultrasonic vibration settings. In this context, we [18] established a molecular dynamics simulation method to reveal the molecular mechanism of interfacial friction heating under ultrasonic vibration conditions, and concluded that the concentrated high frequency chain motion caused by reciprocal sliding friction has a higher heating capability than traditional elastoplastic deformation caused by single sliding friction. The bond length and angle, as well as the dihedral angle, of the polymer macromolecular chain have a significant impact on the evolution of the friction energy.

It is worth noting that, in addition to the homogeneous polymer–polymer interfacial friction mentioned above in the UPMIM equipment’s plasticizing unit, there is also polymer–metal heterogeneous interfacial friction found in other places, such as the interface between plastic pellets and the sonotrode’s end surface, as well as the barrel surface. Because metals and polymers have vastly different thermodynamic characteristics [19,20,21], polymer–metal heterogeneous interfacial friction is expected to behave very differently from homogeneous polymer–polymer interfacial friction. However, relatively few studies on polymer–metal interface friction under the action of ultrasound is currently available. For example, Candice et al. [22] used an experimental analysis of ultrasonic welding between polymers and metals, and concluded that the lap of heterogeneous interfaces can achieve greater shear strength. Dai et al. [23]studied the friction characteristics of two interface structures of lubricating coatings through all-atom molecular dynamics, and concluded that the kinematic characteristics are affected by adhesion, which the non-uniform friction force is affected by changes in material stiffness and surface roughness under constant pressure. The above conclusions have important guiding significance for the study of heterogeneous interface friction, but most of them are limited to the macroscopic level of cognition.

In this work, MD simulations based on the united-atom model are used to investigate the interface friction mechanism of heterogeneous interface (PE-Ni) in both the SSF and RSF processes. The variety in the mechanical properties of the interface are first qualitatively analyzed in two modes. Second, the dynamic response of friction to the microscopic variables of the coupling interface is studied, such as molecular chain conformation, interface diffusion, and internal energy changes. Then, the primary factors of frictional heating at the heterogeneous interface are discussed, for which the interface temperature distribution and heating rate are used as indicators. The heat generation mechanism of the interface under ultrasonic conditions are discussed further below.

## 2. Results and Discussion

### 2.1. Interfacial Loading Characteristics under Precompaction

The initial loading pressure was adjusted and the interface temperature was analyzed. As seen in Figure 1a, under load pressures of 0.5 MPa, 0.9 MPa and 1.2 MPa, the interface temperature peaks are shown at t = 0.38 ns, t = 0.40 ns and t = 0.42 ns, respectively. Therefore, the interfacial heating rate increases with increasing loading pressure. However, the equilibrium temperature is not affected by the loading pressure. This is because the movement of molecular segments towards the interface is aggravated by the increase in pressure, while the increase in molecular velocity at the microscopic level is manifested as the increase in temperature at the macroscopic level. Afterwards, the temperature drops as the molecular thermal motion is limited by the interfacial barrier. Compared with the loading temperature rise characteristic of the polymer–polymer homogeneous interface [18], the temperature rise in the PE/Ni heterointerface is relatively low, which may be caused by two reasons. On the one hand, the thermal conductivity of the metal contact interface is higher (about 300 times) than that of the polymer interface [24,25,26], and the interface heat will diffuse through the metal layer. On the other hand, because there is no entanglement of molecular chains at the heterointerface, the effect of adhesion is weak, so the influence on the movement of molecular chains is not obvious [27].

The loading process of the system can be divided into three states: separation, contact and equilibrium. In a separation state, there is no energy transfer in the interface. In the contact state, the microscopic interface is locally deformed, the temperature rises, the molecular motion is active, and the interface is coupled. In the equilibrium state, the system energy becomes stable and a new contact interface is formed. As shown in Figure 1b, in the process from t = 0 ns to t = 0.31 ns, the surface atoms will transition from constant acceleration to variable acceleration (a < 0) at the initial position of 10 Å. At the initial temperature of 300 K, the amorphous polyethylene block is penetrated by the Ni interface to a depth of 3.6 Å, which transitions from the initial contact stage to the balance stage. Therefore, it is concluded that Ni penetrates through the PE interface to form a new mixed layer, and the hetero interface is in a dynamically stable state.

The variation trend of the contact temperature at the PE/Ni interface is quite different in the vertical and horizontal distributions, as shown in Figure 2a. The transverse horizontal temperature is named the interface temperature, which reflects the local reaction under the same plane as a reference for other temperature points on the same plane. Instead, the longitudinal temperature reflects the uniformity of temperature diffusion and temperature rise. During the loading process, even if Ni metal shows high thermal conductivity, it will still trigger heat accumulation on the PE interface in a short period of time. At t = 0.308 ns, due to differences in heat storage and thermal conductivity [28,29], the temperature distributions across horizontal and vertical interfaces are distinct. The steady state values (316.5 K, and 313.4 K) of the two are also different in the balance state. The interface and vertical temperature distribution clouds at t = 0.308 ns are shown on the right side of Figure 2b. Given that the heat distribution on the polyethylene interface is more uniform than on the metal, the basic temperature of the interface will be maintained at approximately 340 K. The difference in the longitudinal temperature gradient of the PE–Ni system is indicated by the thermal uniform thicknesses of Δx1 and Δx2, with the results showing that the heat transfer capacity of polyethylene along the temperature gradient is lower than that of metal, but its interface is characterized by a higher transient heat generation ability due to the large elastic deformation. In addition, polyethylene is still solid in the loading equilibrium stage, and the heat generated by viscoelastic interactions can be ignored in the sliding friction stage.

### 2.2. Interfacial Friction Characteristics under Sliding Friction

#### 2.2.1. Interfacial Contact Mechanics

The rearrangement of molecular chains at the interface at the microscopic scale is affected by friction [30,31,32]. The action form of the friction force in the PE/Ni sliding process can be divided into two parts: (1) The elastic deformation of the interface and the lateral slip of the wave crest form the plough force in the horizontal direction; (2) Interface coupling, polyethylene molecules adhere to the nickel layer, and shear sliding occurs [33]. Affected by the interface geometry, the distributions of adhesion and ploughing forces are different, as shown in Figure 3a. The plough force is mainly concentrated in the vicinity of the crest and the trough, while the sidewalls of the convex body are acted on by adhesion. The force analysis of the contact point during SSF and RSF is illustrated in Figure 3b. The results show that the SSF friction force is dominated by the plough force, the RSF contact point moves down, and adhesion force plays a leading role.

The friction force of the nanoscale friction interface is dependent on speed, and the change in speed follows the law of dynamic friction [34]. This study explores the microscale interfacial friction change law under SSF and RSF through speed control. Figure 4 illustrates the change trend of the friction force of the interface in both the RSF and SSF modes under constant pressure, which is used to qualitatively characterize the mechanical properties of the interface in the plasticization stage. Secondly, the average friction force is used as an indicator to quantitatively measure the friction performance with different loads. The values of nanoscale interface friction forces always vary nonlinearly. In a motion system, in the time-varying law of forces of particles in the interface layer along the X, Y and Z directions, the interface friction force $F_{f}$ can be calculated using the following formula:(1)$$F_{f}=\sqrt{F_{x}{}^{2}+F_{y}{}^{2}+F_{z}{}^{2}}$$

Figure 4a,c,e shows that the friction forces of the SSF and RSF change steadily after t = 0. 470 ns, t = 1.173 ns and t = 1.575 ns, respectively. During the SSF process, the friction force decreases with increasing sliding speed in the initial stage. However, in the stable phase, the friction force is not significantly affected by the sliding speed [35]. During the RSF process, the friction force generally increases with the amplitude and frequency. An increase in amplitude results in a steadier increase in friction compared to frequency. Figure 4b shows that, during the SSF process, when the load pressure is at 1.2 MPa, the effect of velocity on the average friction transition will change from a positive to a negative correlation. In the initial stage of sliding friction, the interfacial friction characteristics are changed by increasing the pressure. Figure 4d,f shows that the increase in load pressure during RSF suppresses the dependence of frequency and amplitude on interfacial friction [36]. The effect of pressure is most pronounced at variable frequencies.

#### 2.2.2. Polymer Segment Rearrangement

In the friction system, the morphology (microconformation and macroconformation) of the polyethylene chain changes as a result of friction (adhesion and plow effects) [16]. During the compression process, the molecular chains move regularly and the temperature increases. According to the principle of entropy increase, the polyethylene molecular chain transitions from regular to random motion, and the groups (CH3–CH2) rotate more freely. In the process of molecular chain configuration change and diffusion, a new interface coupling is formed [37,38,39], as shown in Figure 5. In addition, the polyethylene molecular chain becomes curled and shows good flexibility. To further characterize the influence of SSF and RSF interface friction on molecules, two sets of simulations were carried out on the influence of configuration: (1) a single movement of the nickel metal block is made at a speed of 20 m/s, to characterize the molecular configuration in the shift from t = 0.30 ns to t = 0.65 ns, and (2) the nickel metal block reciprocates at 80 GHz and 0.5 nm, to characterize the molecular configuration in the shift from t = 1.0 ns to t = 1.6 ns.

As seen in Figure 6a, in the SSF process, the atomic diffusion distribution at the interface expands parabolically along the velocity direction, starting from the initial vertical state. However, in the RSF process, due to the sinusoidal motion in the period of T = 1.42 ns, the molecular chain reciprocates. Due to the thermal motion of molecular chains, the mechanical interlocking ability of the heterointerface decreases at t = 0.52 ns and t = 1.52 ns, respectively. After slippage occurs at the interface, the molecular chains exhibit parabolic and symmetric diffusion in the Y direction during the SSF and RSF processes, respectively. Different from the homogeneous interface friction, the interfacial molecular chain entanglement ability is lower, and the molecular chain rearrangement is relatively less active [18]. As observed in Figure 6b, the molecular chain configuration can change in the interface between the SSF and RSF. The molecular chains of Zones 1 and 2 are characterized for different stages, with the results showing that SSF focuses on the shearing and stretching of the chains. RSF drives the interface molecular chains to be repeatedly arranged in period T, and the transverse shear strength of the molecular chains is low. The resulting molecular chain finally falls in a diffusion state due to the shear stretching of the chain and the hike of the interface temperature.

From the above visual angle, morphological changes in the interface molecular chains are observed. In three-dimensional space, the molecular chains evolve in any direction. Due to the effect of friction and heat, the final shape of the molecular chains diffuses. For the accurate analysis of the evolution of interface molecular chains under ultrasonic action, the interfacial diffusion characteristics were explored by calculating the MSD mean square displacement curve of the molecular chains. By linearly fitting the data points from t = 0.3 ns to t = 1.6 ns, according to the Einstein relationship, the diffusion coefficient is calculated by 3 times the slope [40,41]. To study the diffusion of molecular chains by characterizing the diffusion coefficient of interfacial, four sets of simulations were carried out by adjusting the motion parameters of the metal layer: (1) Single sliding friction, different speeds (10 m/s, 15 m/s, 20 m/s); (2) Single sliding friction, different loads (0.5 MPa); (3) Reciprocating sliding friction, same amplitude (0.5 nm), different frequencies (60 GHz, 70 GHz, 80 GHz); (4) Reciprocating sliding friction, natural frequency (80 GHz), and different amplitudes (0.5 nm, 1.0 nm, 1.5 nm).

Figure 7a,b shows that the pressure and velocity have a nonlinear correlation with the displacement of interfacial molecular chain during the SSF process. The MSD mean square displacement of the molecular chain increases with increasing velocity, while increasing pressure decreases it. The diffusion rate of interfacial molecules increases first and then tends to be smooth. It can be seen from Figure 7c,d that the frequency and amplitude of the RSF process cause the irregular and rapidly diffusion of the interface, and the overall trend will first rise sharply and then drop to a steady state. Based on the above analysis, the comparative analysis of the SSF and RSF processes shows that a single sliding friction has a significant effect on molecular chain diffusion. The smaller the pressure is, the more intense the molecular chain diffusion, and the increase in speed will accelerate the diffusion performance. In the RSF process, the diffusion properties of the molecular chains are enhanced with increasing the frequency and amplitude [18]. The MSD displacement curve drops sharply after reaching a peak and eventually tends to be smooth. This phenomenon, as shown in Figure 6a above, is attributed to the disentanglement of molecular chains affected by temperature and the weakening of the interfacial coupling effect. Meanwhile, it is also verified that the molecular chain mobility is higher in the initial stage of friction, and the equilibrium stage is in a steady state.

#### 2.2.3. Frictional Energy Transfer

The interaction between microscopic interfaces can be simply regarded as frictional work, part of which is used to overcome interface deformation. Due to the study of the contact between the metal and the polymer, the deformation is mainly reflected in the configuration change of the polymer molecular chain; the other part relies on the rearrangement of the molecular chain to output heat and achieve interface coupling in the steady state. Interfacial coupling occurs during both loading and friction processes. The heterogeneous interface PE/Ni conducts with the ΔT/t temperature gradient to achieve an interface steady state, as shown in Figure 8a. The change in the interface heat in the friction process can be characterized by the change in the temperature of the contact area. The heat transfer capacity of the interface is related to the thickness of the material and the thermal conductivity. The study of the temperature rise characteristics of the constant thickness interface layer mainly considers the difference in thermal conductivity. For the analysis of the heat of the nickel layer and the polymer layer, we first considered the changing trend of the thermal conductivity of the material with temperatures. Figure 8b shows that the thermal conductivity of nickel and polymer changes with temperature, which is adapted from it [28]. The thermal conductivities of the metal and polymer are very different [20], which results in different temperature characteristics of the interface friction between the two modes. As shown in Figure 8c, the heat generation characteristics of the PE–metal interface in the SSF and RSF modes were studied. The results show that, compared with SSF, RSF has a higher temperature rise due to the high frequency action of molecular chains [18]. Even if there are differences in thermophysical properties at the hetero interface, the transient temperature rise will still cause heat accumulation at the PE interface with a temperature difference of around 35 K.

Based on the above analysis, frictional heat is mainly transmitted through the metal layer. Due to the relatively high sliding speed of the interface, a large amount of heat is transferred from the metal layer, which still causes heat accumulation at the polyethylene interface. To further analyze the heat generation mechanism, the temperature change curve of the contact layer and the average temperature rise rate were used to characterize the effect of ultrasonic parameters on the temperature rise of the interface. Meanwhile, the maximum temperature rises of the curve at different times was used to characterize the interface temperature contours.

From Figure 9a,c,e, it can be seen that the sliding temperature rise characteristics show a trend of first increasing and then gradually decreasing to a steady state. SSF and RSF have great differences in the temperature rise changes of the interface. In the SSF process, the speed increases, and the interface temperature and the temperature rise rate increase. Figure 9b shows that an increase in load increases the temperature rise rate. In the RSF process, the amplitude and frequency are positively correlated with the temperature rise characteristics, but the interface heat generation is greatly affected by the amplitude. Figure 9d reveals that, during the frequency conversion friction process, the increase in the load will reduce the temperature rise rate under the action of frequency conversion friction. Figure 9f shows that during the variable amplitude friction process, the increase in load increases the temperature rise rate, but the effect is not evident. According to the evolution law of temperature rise of RSF and SSF, the temperature change depends on the work performed by the friction force on the molecular chain. The change of the above parameters can actually be regarded as the regulation of the average rate, and the temperature rise rate depends on the effective work.

During the friction process, the friction interface is in a state of squeezing, which limits the active space of the molecular chain. Sliding friction periodically changes the spatial configuration of the molecular chain, and the heat generated by high-speed friction increases sharply, resulting in continuous changes in the bond length, bond angle and dihedral energy of the molecular chain. To further study the relationship between the temperature and the energy conversion form of the friction interface, the influence of ultrasonic plasticization on the energy of the coupling interface was qualitatively analyzed by adjusting the speed and amplitude of the slider.

Figure 10 shows the change trend of the PE/Ni interface energy during the SSF and RSF processes. The temperature rise trend is a macroscopic characterization of the total energy change ∆*E* ($E-E_{O}$), while the law of energy evolution at the microscopic scale still needs to be further explored. Under the action of the two modes, we can see that the dihedral energy, van der Waals energy and kinetic energy are the most evident changes in the energy curve. This result is attributed to the interfacial molecular chain rearrangement, which is consistent with the result of Qiang et al. [18]. Figure 10a,c,e reflects the influence of speed on the interface energy in the SSF process. The total energy increases first and then stabilizes. An increase in speed increases the dihedral energy conversion rate and promotes the conversion of kinetic energy to potential energy. Figure 10b,d,f reflects the evolution of the amplitude of the interface energy in the RSF process. The interface energy change rate is relatively block, similar to its temperature rise characteristics. As the amplitude increases, the dihedral angle energy change decreases, the potential energy is transformed into kinetic energy, and the rate of change of kinetic energy increases.

## 3. Materials and Methods

### 3.1. Material

Polyethylene was chosen to study the polymer–metal friction mechanism in this paper due to its better physical and mechanical properties [42,43]. In terms of metals, nickel is always treated as a wear-resistant and antioxidant material to plate on the surface of the ultrasonic tool head in the plasticizing process [44,45]. In the friction analysis of polyethylene and Ni metal, first, the glass transition temperature of the polyethylene material is determined by the intersection of the volume ratio and the slope of the temperature curve, and the optimal temperature for the material research is selected. In the initial phase, the polyethylene system was relaxed at 500 K in the NVT (the atomic number N, volume V and temperature T remain unchanged) and compressed to a specified density in the NPT (constant-pressure, constant-temperature) ensemble to achieve equilibrium. After that, the system temperature was lowered to 100 K, and the volume changes of polyethylene were collected every 50 timesteps. Figure 11a shows the density as a function of temperature. According to the curve fitting result of the image, Tg = 288.6 K, which is similar to the results of previous research in the literature [46]. This provides critical conditions for the next step to study the tribological properties by setting different temperatures according to the glass transition temperature.

After balancing the amorphous polyethylene, the bond length, bond angle and dihedral angle of the polyethylene molecular chain were analyzed by statistical methods. Figure 11b–d are examples of the bond length, bond angle, and dihedral angle distribution after the system was balanced at a temperature of 100 K. The bond length and bond angle were normally distributed. After the system was balanced, the bond length and bond angle were 110.7° and 1.535 Å, respectively, which are slightly higher than the values of potential parameters $\theta_{0}$ and $r_{0}$. Due to the relatively large molecular weight of the system, the parameter deviation was within a reasonable range, indicating that the molecular model was suitable for further investigation.

### 3.2. Methods

#### 3.2.1. Initial Model Analysis

Figure 12 shows the analysis of the polymer–metal boundary conditions at different stages of the ultrasonic plasticization process. The process can be divided into two stages: the first stage is the initial loading and plasticized adhesion friction, corresponding to Figure 12a,b. Firstly, polyethylene particles underwent elastic deformation under a constant force Fb. The sideslip displacement between particles was different by Δx1 and Δx2, and the research object can be regarded as the displacement Δx1 + Δx2 under the single sliding friction (SSF) of constant force Fa. Secondly, plasticized adhesion occurred under the action of high-frequency variable force. The particles were heated and adhered, and the volume ratio of the particles to the cavity was obtained. According to the change in ultrasonic plasticizing load characteristics, the axial force Fb and the traction force Fa were approximated as sinusoidal forces with a phase difference of φ, leading to the effect that the bonded atomic layer exhibited reciprocating sliding friction (RSF) with a displacement of Δx.

The PE–Metal hybrid model used Moltemplate and LAMMPS for joint modeling and operation processing, and the results were visualized and analyzed through Ovito. In terms of model structure, the polyethylene layer was composed of a fixed layer, a quench layer and a free layer (total 252 chains; 25,200 monomers). The metal layer was composed of a rigid layer, a metal quench layer and a slide layer. The size of the polyethylene block model in the X–Y–Z direction was 6.4 nm × 6.4 nm× 7.0 nm, the size of the metal Ni model in the X–Y–Z direction was 6.4 nm × 6.4 nm × 3.6 nm, and the model gap was set as 1.0 nm. The three-dimensional simulation model is shown in Figure 13. Moreover, the whole simulation process used the Langevin algorithm to control the constant temperature of the Quench layer to 300 K. The temperature of the Slide layer and the Free layer were tested under the micro-regular ensemble (NVE). To ensure the stability of the equation of motion solution, the selected time step was 0.1 Ps. The first 3000 ps was the equilibrium stage, and the NVT ensemble was selected. Then, 6000 ps was the simulation stage, and the NVE ensemble was selected to calculate the friction characteristics of the heterogeneous interface. For the model initial coordinates prepared, the CH2–CH3 monomers were arrayed with initial coordinates (0, 0, 0.36) along the normal vector $\vec{n}$(0, 0, 1) to satisfy the molecular chain with a degree of polymerization of 100. We used a three-dimensional array to fill a rectangular box along the X, Y, and Z directions with a molecular weight of 252. Because of the amorphous distribution of the molecular chains along the Z axis, the uneven color distribution of the surface atoms was used to characterize them to facilitate the study of the state changes of the molecular chains. The X and Y directions were set as periodic boundary, and the Z direction was set as contractive boundaries. The upper and lower walls were arranged along the z-axis, and the energy minimization method was used to compress the polyethylene to the specified density (reference value is 0.91 [g/cm]^3^) [18].To reduce the steady state time, the temperature of the system was reduced from 500 K to 100 K, and the polymer was annealed.

#### 3.2.2. Calculation of the Interaction Potentials and Kinetic Energy 

The interaction potential is the effective interaction between atoms in the force field. The development of the interaction potential affects the application of molecular dynamics in research to a certain extent. The acquisition of its related data was mainly conducted through experimental fitting and semiempirical solution, using the potential energy function to obtain the mechanical properties of the molecules and to obtain the system energy. Its accuracy affects the accuracy of the simulation results. For polyethylene, we referred to the research of D. Hossain et al. on the polymer deformation mechanism [47]. The force field parameters were selected, as shown in Table 1.

The description of the force field under the molecular system was divided into bond interactions and intermolecular interactions (van der Waals forces). For metal modeling, according to the specific characteristics of the research material, there was no obvious bond orientation between atoms, so the metal used the EAM potential to embed the atom [48,49]. This method passed various operational tests, such as migration energy, surface energy, and metal surface geometry. The parameters in the interaction potential between different pairs of metals and polymers adopt the Lorentz–Berthelot mixing rule, in which the energy and molecular size parameters are assumed to be geometric averages and arithmetic averages, respectively. The specific formula is as follows:(2)$$\sigma_{ij}=\frac{\sigma_{ii+}\sigma_{jj}}{2}$$
(3)$$\varepsilon_{ij}=\sqrt{\varepsilon_{ii}\varepsilon_{jj}}$$

The energy characterization is mainly reflected in the energy (kinetic energy $E_{k}$) of the thermal motion of molecules and the energy (potential energy $E_{P}$) exhibited by the intermolecular forces. In the MD simulation, the potential energy was described by analyzing the empirical potential function. The potential function of the system can be decomposed into the joint action of multiple potential functions, and the bond expansion energy, bond bending energy, dihedral twisting energy, and van der Waals action energy are quantified for characterization. The bond interaction includes the kinetic energy of the stretching vibration of the bond, the kinetic energy of the bending vibration of the bond, and the kinetic energy of dihedral torsion. Through the comparison of the three energies, the conformation and energy of the molecule were further optimized. The specific calculation formula is as follows:(4)$$E_{bond}\left(r\right)=\frac{1}{2}K_{b}{\left(r-r_{0}\right)}^{2}\;$$
(5)$$E_{angle}\left(\theta \right)=\frac{1}{2}K_{\theta }{\left(\theta -\theta_{0}\right)}^{2}\;$$
(6)$$E_{dihedral}\left(\varphi \right)=\sum_{i=0}^{3}C_{i}{\left(\cos\varphi \right)}^{i}\;$$
where $K_{b}$ and $K_{\theta }$ represent the bond length and angular force constant, respectively. $r_{0}$ and $\theta_{0}$ refer to the equilibrium bond length and angle, respectively. $C_{i}$ is the dihedral angle polyharmony coefficient; *r* is the distance between two atoms; and the interaction force between molecules is mainly represented by van der Waals forces. The expression is as follows:(7)$$E_{i-j}\left(r\right)=4\varepsilon \left[{\left(\frac{\sigma }{r}\right)}^{12}-{\left(\frac{\sigma }{r}\right)}^{6}\right]+C\;$$
(8)$$C=-4\varepsilon \left[{\left(\frac{\sigma }{r_{c}}\right)}^{12}-{\left(\frac{\sigma }{r_{c}}\right)}^{6}\right]\;$$
where $E_{l-j}\left(r\right)$ represents the interaction energy given by the Lennard-Jones potential. σ is the distance when the potential between atoms is zero, and *C* is the energy displacement coefficient to ensure that the potential energy is continuous at $r_{c}$. In this work, the LJ parameters were $\varepsilon_{\left(Ni-c\right)}=0.0502867ev$, $\sigma_{\left(Ni-C\right)}=3.1454\;$Å, and the cutoff was $r_{c}=2.5\sigma_{Ni-c}$.

The total kinetic energy ($E_{k}$) of the atom group is customized by collecting the average kinetic energy component of each atom.
(9)$${\left(E_{k}\right)}_{tol}=\frac{1}{2}\sum m_{i}\left(v_{ix}{}^{2}+v_{iy}{}^{2}+v_{iz}{}^{2}\right)\;$$
(10)$${\left(E_{k}\right)}_{com}=\frac{1}{2}\sum m_{i}\left({\bar{v_{ix}}}^{2}+{\bar{v_{iy}}}^{2}+{\bar{v_{iz}}}^{2}\right)$$
(11)$$\bar{v_{x}}=\frac{1}{k}\sum \bar{v_{ix}}$$
(12)$$\bar{v_{x}}=\frac{1}{k}\sum \bar{v_{ix}}\;$$
(13)$$\bar{v_{x}}=\frac{1}{k}\sum \bar{v_{ix}}$$
(14)$${\left(E_{K}\right)}_{j}=\frac{1}{n}\left[{\left(E_{K}\right)}_{tot}-{\left(E_{K}\right)}_{com}\right]\;$$
(15)$$\;T_{j}=\frac{2{\left(E_{K}\right)}_{j}}{3k_{B}}\;$$
where ${\left(E_{k}\right)}_{tol}$ stands for total kinetic energy, ${\left(E_{k}\right)}_{com}$ stands for kinetic energy at the center of mass, *i* stands for atomic number, and *k* stands for the number of atoms in the group. Equation (15) is the conversion relationship between energy and temperature.

#### 3.2.3. Calculation Scheme

The friction simulation of the ultrasonic plasticized interface was divided into three parts: load balance, SSF, and RSF. In the initial stage, the normal load $\vec{P}$ (0.5 MPa, 0.9 MPa, 1.2 MPa) was applied to the contact interface, with the glass transition temperature as the limit. The rationality of the model was verified by the interface displacement and temperature rise change, and it was also used as the initial condition of the friction model. Due to the temporal and spatial differences between the micro-nanoscale and macro-scale, the settings of physical parameters (such as velocity, frequency) will be much higher, even in orders of magnitude [50,51,52]. Therefore, the speed along the (0, 1, 0) direction under the SSF mode was set to 10 m/s, 15 m/s, and 20 m/s. In RSF mode, the interface moved along the (0, ±1, 0) direction with a sinusoidal change S = Asin (2πft), where the amplitude A (0.5, 1.0 and 1.5 nm) and the frequency f (60, 70 and 80 GHz). The inner relationship between molecular chain rearrangement (molecular chain configuration change and MSD displacement) and interface friction under two modes was studied. The mechanism of interfacial friction heat generation and system energy conversion was characterized by the calculation of intramolecular energy.

## 4. Conclusions

In this paper, polymer–metal interfacial friction characteristics under ultrasonic plasticizing conditions were investigated by united-atom molecular dynamics (UAMD) simulation. On the one hand, the initial loading characteristics of the interfacial friction model were characterized by the temperature equilibrium and the interpenetration depth of the heterogeneous interface. On the other hand, the interfacial friction characteristics were studied in terms of contact mechanics, polymer segment rearrangement and friction energy transfer. The key findings of this research include the following:

(1) During the initial loading process, the formed mixed heterogeneous interfacial layer is in a dynamic stable state. The interfacial heating rate increases with increasing loading pressure. Largely deformed polymer blocks exhibit higher transient temperatures and heat build-up. This difference in the instantaneous temperature change between the polymer and the metal bulk is thought to be related to the different thermal properties of the heterointerface.

(2) For the interfacial contact mechanics in the sliding friction process, the initial interfacial friction force in the single sliding friction (SSF) process decreases significantly with the increase in the sliding velocity. However, the increase in amplitude results in a more stable increase in RSF interfacial friction compared to frequency. The effects of these parameters on friction in SSF and RSF weakened with increasing loading pressure, which proved to be the parameter with the greatest effect on interfacial friction.

(3) In terms of the rearrangement of polymer segments, new interfacial couplings are formed in the case of heterointerface molecular chain configuration changes and diffusion. However, single-slip friction would be more favorable for interfacial diffusion at lower pressures with increased velocity. Compared with the increase in the frequency, the increase in the amplitude further promotes the rearrangement of the molecular chain and enhances the effect of interfacial plasticization.

(4) In terms of frictional energy transfer, the interface temperature rise characteristic is used as a macroscopic characterization of energy change. The difference in temperature rise characteristics between SSF and RSF modes is mainly manifested in the temperature rise rate. An increase in load results in an increase in the rate of temperature rise, where the interfacial heating is more affected by the magnitude. Contrary to the effect of the amplitude effect, the increase in speed promotes the energy transformation of the dihedron, and the kinetic energy is transformed into potential energy.

## Figures and Tables

**Figure 1 ijms-23-02829-f001:**
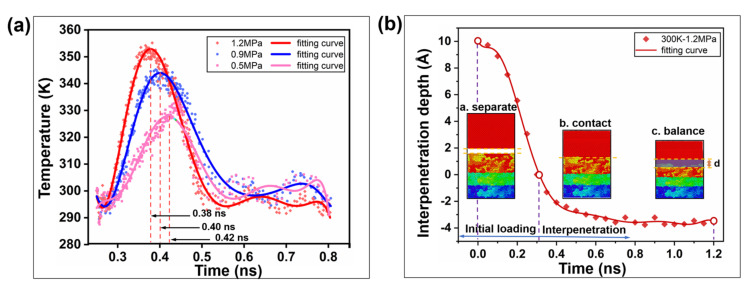
The state change of the PE/Ni interface under load pressure. (**a**) The variation trend of interface temperature under loading pressure (0.5 MPa, 0.9 MPa, 1.2 MPa). (**b**) Dynamic response of the loading properties of heterointerfaces with time at 300 K. Model snapshots represent three stages in the loading process. The yellow dashed lines represent the PE bulk and Ni contact boundaries. Light red, green, and blue represent the free layer, the quenched layer, and the fixed layer, respectively, which form an interfacial mixed layer in the balance stage.

**Figure 2 ijms-23-02829-f002:**
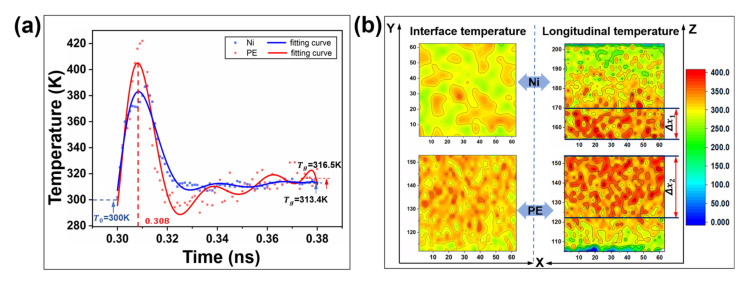
Temperature characteristics at the heterogeneous interface between Ni and polyethylene. (**a**) Variation trend of the average temperature of the PE–Ni interface with time; (**b**) Temperature distribution cloud map of the heterointerface.

**Figure 3 ijms-23-02829-f003:**
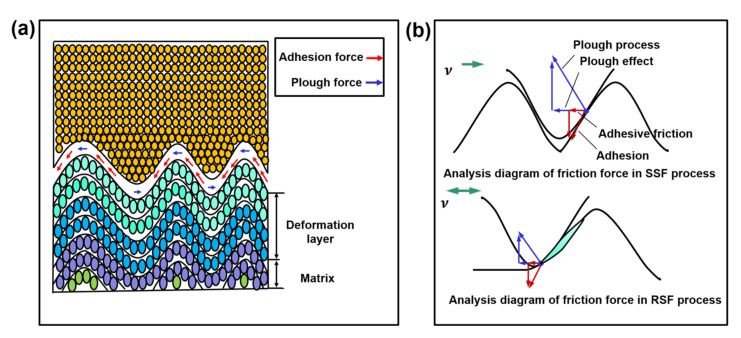
Mechanical analysis of the friction model. (**a**) Assumption of the interface structure of the friction microcontact. (**b**) Mechanical analysis of the friction model structure.

**Figure 4 ijms-23-02829-f004:**
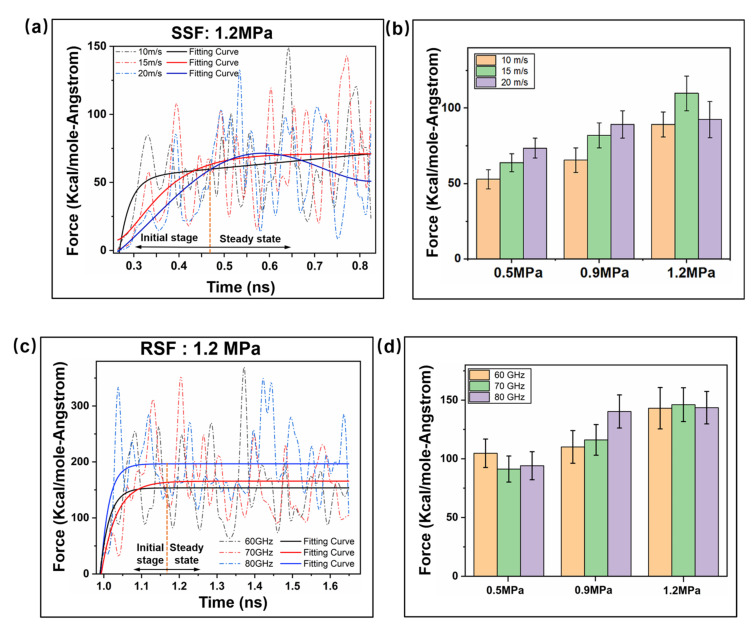
Variation in the interface mechanical properties between the SSF and RSF. (**a**,**c**,**e**) Velocity (10 m/s, 15 m/s, 20 m/s), frequency (60 GHz, 70 GHz, 80 GHz), and amplitude (0.5 nm, 1.0 nm, 1.5 nm) on the friction force of the heterogeneous interface impact. (**b**,**d**,**f**) The average mechanical parameters in the two modes are affected by the load (0.5 MPa, 0.9 MPa, 1.2 MPa).

**Figure 5 ijms-23-02829-f005:**
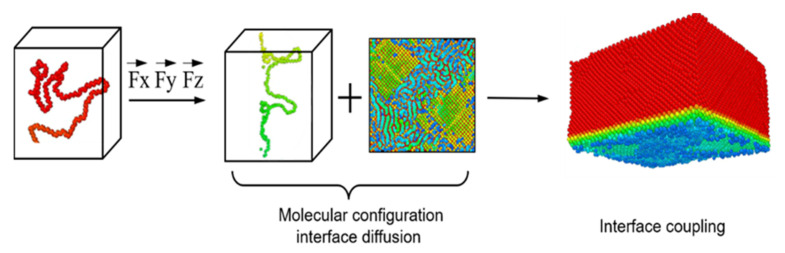
PE–Ni interface coupling.

**Figure 6 ijms-23-02829-f006:**
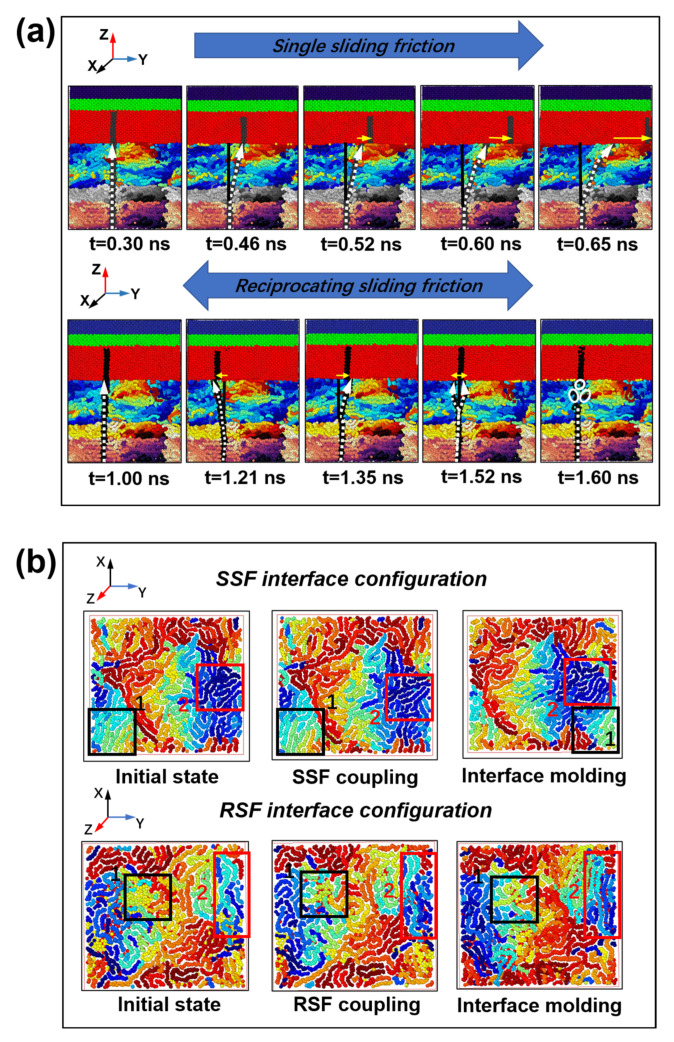
The evolution of the vertical and horizontal structure of the polyethylene interface during the SSF and RSF processes. (**a**) The friction interface diffuses longitudinally in the process of SSF and RSF. (**b**) The molecular chain configuration changes at the coupling interface. White atoms are the selected atoms to characterize molecular diffusion. The horizontal arrow is above the free layer and characterizes the slip distance.

**Figure 7 ijms-23-02829-f007:**
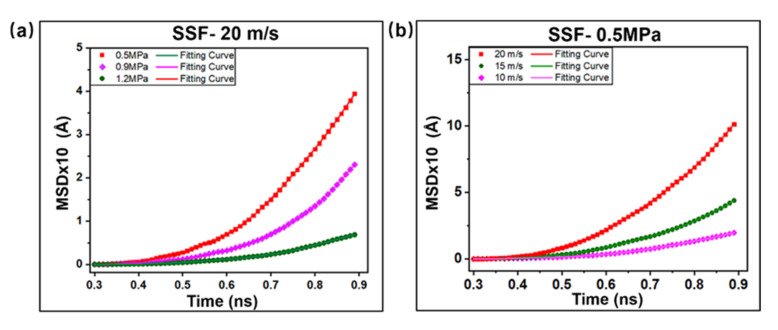
Analysis of polyethylene operation results in Z-axis MSD under different modes. (**a**) Single sliding at different speeds (10 m/s, 15 m/s, 20 m/s). (**b**) Single sliding under different loads (0.5 MPa, 0.9 MPa, 1.2 MPa). (**c**) Reciprocating sliding friction (RSF) at different frequencies (60 GHz, 70 GHz, 80 GHz). (**d**) Reciprocating sliding friction (RSF) at different amplitudes (0.5 nm, 1.0 nm, and 1.5 nm).

**Figure 8 ijms-23-02829-f008:**
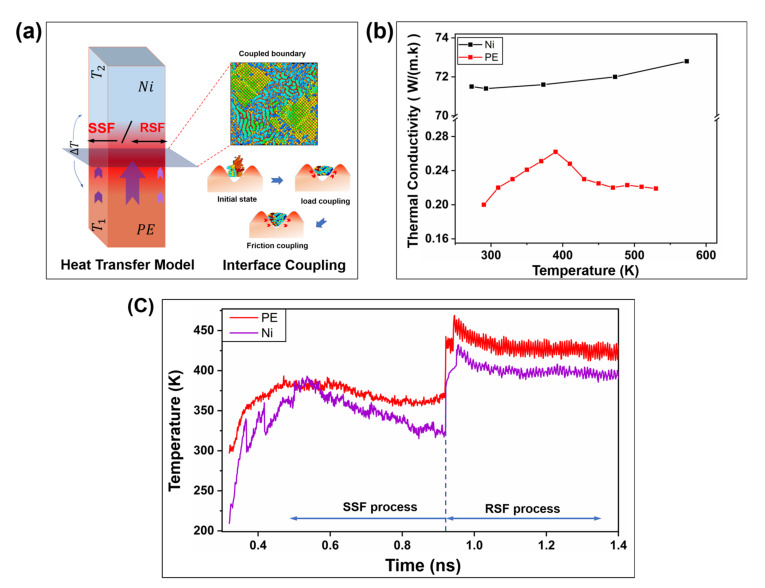
The change in the thermal conductivity of nickel and polymer and the interface heat transfer in SSF and RSF modes. (**a**) Schematic diagram of the contact morphology of the coupling interface. (**b**) Changes in the thermal conductivity of nickel and polymer. (**c**) Interface heat transfer characteristics during plasticization.

**Figure 9 ijms-23-02829-f009:**
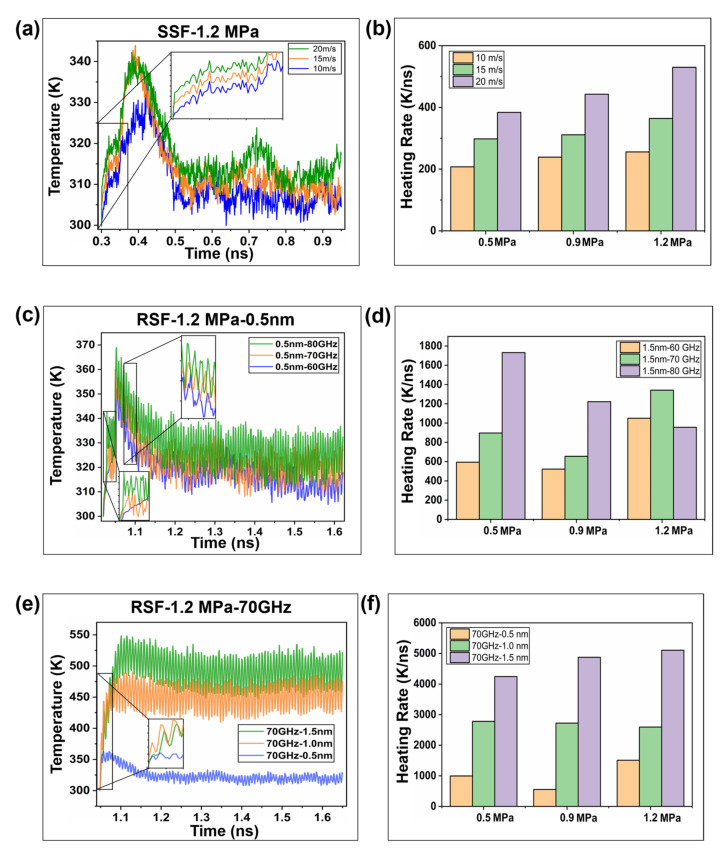
(**a**,**c**,**e**) The change trends of the interface temperature under different sliding speeds (10 m/s, 15 m/s, and 20 m/s), different frequencies (60 GHz, 70 GHz, and 80 GHz), and different amplitudes (0.5 nm, 1.0 nm, and 1.5 nm) of a single sliding friction (SSF) and reciprocating sliding friction (SSF), respectively. (**b**,**d**,**f**) The effects of different loads (0.5 MPa, 0.9 MPa, and 1.2 MPa) on the interface temperature rise rate in the two modes.

**Figure 10 ijms-23-02829-f010:**
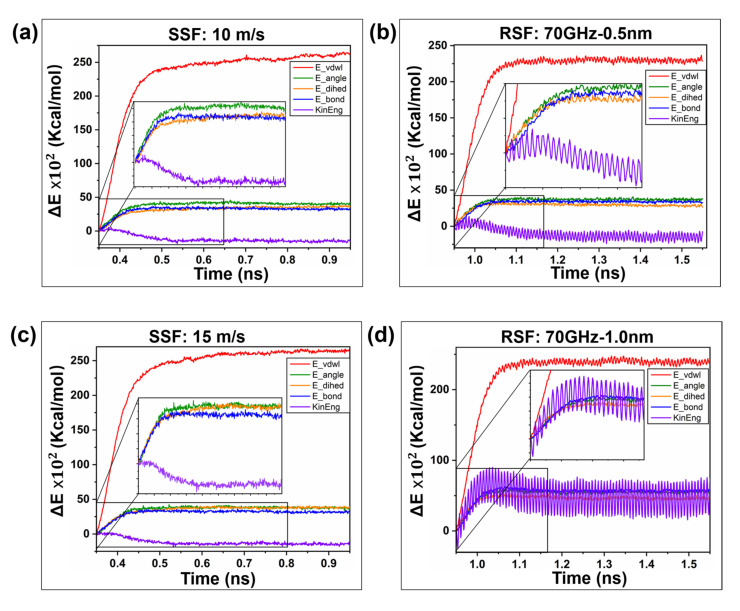
Energy analysis of a single sliding friction (SSF) and reciprocating sliding friction (RSF) interface. (**a**,**c**,**e**) The effect of speed on the polyethylene bond length, bond angle, dihedral angle and van der Waals energy of a single sliding interface. (**b**,**d**,**f**) The evolution of polyethylene interface energy during reciprocating sliding is regulated by amplitude.

**Figure 11 ijms-23-02829-f011:**
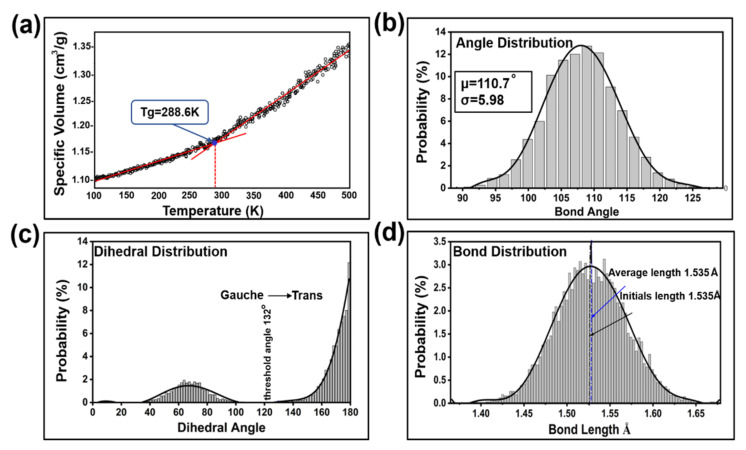
Structure optimization characterization of amorphous polyethylene. (**a**) Variation of specific volume with PE volume temperature. (**b**–**d**) Bond angle, dihedral angle, and bond length distribution in PE block after structure optimization.

**Figure 12 ijms-23-02829-f012:**
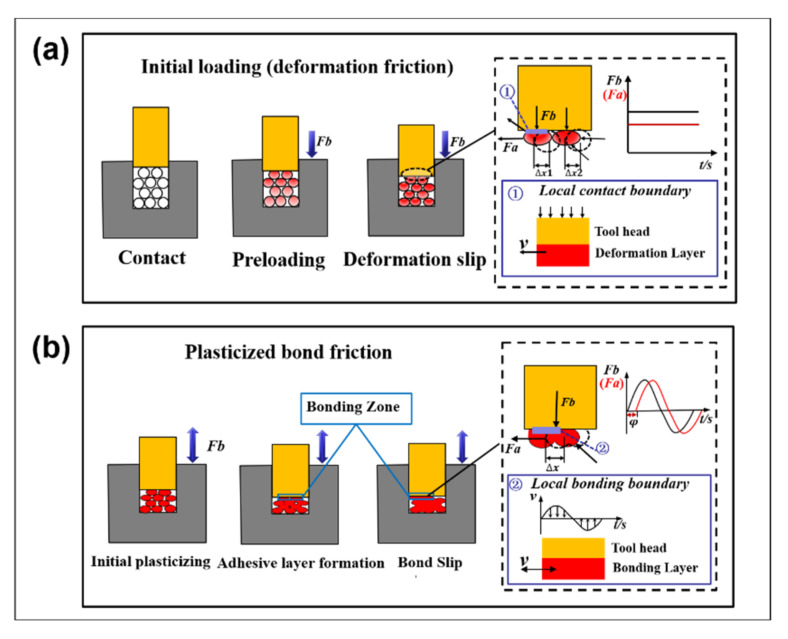
Friction mechanism and force analysis at different stages. (**a**) Single sliding friction (SSF); (**b**) Reciprocating sliding friction (RSF).

**Figure 13 ijms-23-02829-f013:**
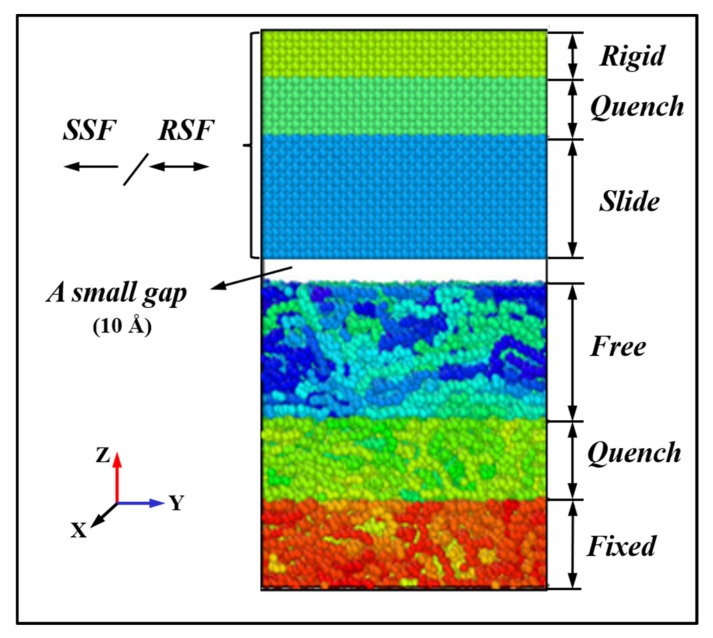
A front view of an amorphous PE/Ni sliding friction hybrid model.

**Table 1 ijms-23-02829-t001:** Force field potential parameters of polyethylene MD simulation calculation.

Interaction	Parameter	
Bond length	$K_{b}=350\;\mathrm{kcal}\cdot {\mathrm{mol}}^{-1}$	$r_{0}=1.53\;$Å
Bond angle	$K_{\theta }=60\;\mathrm{kcal}\cdot {\mathrm{mol}}^{-1}\cdot {\mathrm{rad}}^{-2}$	$\theta_{0}=109{}^{\circ}$ (1.911 rad)
Dihedral angle	$C_{0}=1.736\;\mathrm{kcal}\cdot {\mathrm{mol}}^{-1}$ $C_{2}=0.776\;\mathrm{kcal}\cdot {\mathrm{mol}}^{-1}$	$C_{1}=-4.490\;\mathrm{kcal}\cdot {\mathrm{mol}}^{-1}$ $C_{3}=6.99\;\mathrm{kcal}\cdot {\mathrm{mol}}^{-1}$
Lennard–Jones	σ=4.01 Å	$\varepsilon =1.12\;\mathrm{kcal}\cdot {\mathrm{mol}}^{-1}$

## Data Availability

All data generated or analyzed in the current study are included in this article.

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
