# Peer review of "Polymer–Metal Interfacial Friction Characteristics under Ultrasonic Plasticizing Conditions: A United-Atom Molecular Dynamics Study"

_ijms, 2022, doi:10.3390/ijms23052829_

Round 1

Reviewer 1 Report

The manuscript “Polymer-Metal Interfacial Friction Characteristics under Ultrasonic Plasticizing Conditions: A United-atom Molecular Dynamics Study” by Wangqing Wu, Changsheng He, Yuanbao Qiang, Huajian Peng and Mingyong Zhou is an interesting computational study of mechanistic processes at the molecular level. It certainly describes phenomena important technologically: friction, heat transfer at the interface, etc. I recommend its publication after the issues listed below are resolved.

1. The intended audience: I am not sure that IJMS is the best platform for this manuscript. The Polymers journal would suit better – but this is to be agreed between the authors and the Editorial Office.

2. The manuscript lacks important information related to the computational setup. In particular:

- what software was used to carry out the simulations? Some available code such as GROMACS, LAMMPS, or your own programs?

- what were the protocol details: time step, presence and type of thermostats / barostats etc.?

- how were the initial coordinates prepared? Were the polyethylene chains uniformized, or were their lengths diversified?

3. In section 2.2.2. the authors write “In the compression state, the temperature will rise, and the molecular chain movement is relatively active. According to the principle of increasing entropy, the polyethylene molecular chain will rotate more freely.” Can this be substantiated in the simulations? One could think that the effect of increased temperature might be overcome by the presence of pressure, not allowing the molecules to move freely… Please discuss and comment.

4. There are small presentation and language mistakes, a few of them are listed here:

- lines 220-221: do the authors really mean 60 Hz, 70 Hz, 80 Hz?

- line 311, caption to Figure 10: the plotted values are NOT bond length, angle or dihedral angle, but rather the force field energies related to these parameters.

- line 430: “stiffness coefficient” is routinely called “force constant” within the harmonic approximation used in this study.

- line 470: “unite-atom”

- page 15, equations (9) and (13): the label for “center of mass” is “com” and “cm” – please unify

Reviewer 2 Report

This is interesting manuscript dealing with molecular dynamics simulation of interaction of metal phase with polymer phase under the mechanical force exerted externally. The discussion of results is exhaustive but the description of methodology and relation to macroscopic observables needs extension. Therefore, the manuscript needs revision as described below.

Figure 1. Parts a and b are misplaced

The rise of temperature in Fig.1 is due to applied force, however, to my best knowledge the authors are using standard newton dynamics (not dissipative particle dynamics). Thus, the temperature increase in such a case is a kind of artifact because system is not working in equilibrium with a heat bath. On the other hand if the authors applied any mathematical thermostat then the temperature increase is even more spurious. The authors should address this point carefully.

How they define the “interface temperature” and “longitudinal temperature”?Is there any thermostat or is this the NVE simulation?

Why they apply the coarse grained model of polyethylene? It is not a big deal to use all atom model and standard parameterization like opls or amber.

A closer description of a relation between model predictions and possible experimental observation would be welcome. The authors discuss mainly microscopic parameters (excluding temperature which definition is not obvious, as I mentioned) without relation to any macroscopic observables.

Reviewer 3 Report

Thank you for submitting your paper. The work done here draws attention to a significant subject in polymer-metal Interfacial friction characteristics. I have found the paper to be interesting. However, several issues need to be addressed properly before the paper is being considered for publication. My comments including major and minor concerns are given below:

  1. Please consider reviewing the abstract and highlight the novelty, major findings, and conclusions. I suggest reorganizing the abstract, highlighting the novelties introduced. The abstract should contain answers to the following questions:
  2. What problem was studied and why is it important?
  3. What methods were used?
  4. What conclusions can be drawn from the results? (Please provide specific results and not generic ones).
  5. The abstract must be improved. It does not read well at all. Please use numbers or % terms to clearly shows us the results in your experimental work. Please expand the abstract.
  6. Please consider reporting on studies related to your work from mdpi journals.
  7. The introduction must be expanded, please consider improving the introduction, provide more in-depth critical review about past studies similar to your work, mention what they did and what were their main findings then highlight how does your current study brings new difference to the field.
  8.  
  9. Results and Discussion??? Where is the materials and methods section?
  10. Line 95 by how much it is better? Please support this claim with a reference(s) and also show clearly in % terms how much better it is. Using words such as better or worse is not scientific, please use more appropriate scientific wording such as higher, lower.
  11. Lines 95-96 please discuss this further and support with references, a lot of basic info given but not properly explained.
  12. Please consider enlarging the images in figure 6
  13. Line 233 “and then tends to be gentle” this is not a scientific way to describe an engineering behaviour of materials, please rephrase. Perhaps use the word “smooth”? or something more engineering sound.
  14. Line 234 again “violent diffusion” what does violent mean here? Please rephrase the whole paragraph and make sure to use scientific/engineering sound wording, this issue must be checked elsewhere in the manuscript.
  15. Line 264 “the RSF had dramatic changes in temperature” by how much? Please use numbers or % terms to clearly tell us how dramatic is it! And also remove the word dramatic, use more common words such as significant or similar wording.
  16. Ok now I see materials and method section is section 3, but why? I think the authors should switch the order of section 2 and 3, please check this carefully and update the manuscript.
  17. Please add a table to list all nomenclature, Greek symbols and abbreviations at the end of the manuscript.
  18.  
  19.  
  20. The results are merely described and is limited to comparing the experimental observation and describing results. The authors are encouraged to include a more detailed results and discussion section and critically discuss the observations from this investigation with existing literature.
  21. Conclusion can be shortened or perhaps consider using bullet points (1-2 bullet points) from each of the subsections.
  22. The writing style needs to be checked everywhere in the manuscript, I only highlighted some lines but there are so many which needs rephrasing and improving, please make sure to address this for the 2nd round.

Round 2

Reviewer 2 Report

I recommend the manuscript for publication though I am not fully convinced with the explanation given by authors and related to observation of temperature increase. A lot of things could be discussed....

Reviewer 3 Report

All questions answered and paper can be accepted